# New 11-Methoxymethylgermacranolides from the Whole Plant of *Carpesium divaricatum*

**DOI:** 10.3390/molecules27185991

**Published:** 2022-09-14

**Authors:** Tao Zhang, Haixin Zhang, Chunyu Lin, Lu Fu, Zhongmei Zou

**Affiliations:** 1Institute of Medicinal Plant Development, Chinese Academy of Medical Sciences and Peking Union Medical College, Beijing 100193, China; 2Key Laboratory of Quality Research in Chinese Medicine, Guangdong-Hong Kong Macao Joint Laboratory of Respiratory Infectious Disease, Macau Institute for Applied Research in Medicine and Health, College of Pharmacy, Macau University of Science and Technology, Macau 999078, China

**Keywords:** *Carpesium divaricatum*, methoxymethylgermacranolides, absolute configuration, cytotoxicity, PGE2

## Abstract

Eight new 11-methoxymethylgermacranolides (**1**–**8**) were isolated from the ethanol extract of the whole plant of *Carpesium divaricatum*. The planar structures and relative configurations of the new compounds were determined by detailed spectroscopic analysis. The absolute configuration of **1** was established by electronic circular dichroism (ECD) spectrum and X-ray crystallographic analysis, and the stereochemistry of the new compounds **2**–**8** were determined by similar ECD data with **1**. The absolute configurations of **5** and **7** were further confirmed by using quantum chemical electronic circular dichroism (ECD) calculations. Compound **4** exhibited weak cytotoxicity against MCF-7 cells. Compound **8** could potently decrease PGE2 productions in LPS-induced RAW 264.7 cells.

## 1. Introduction

Sesquiterpenoid lactones have in many instances been instrumental in providing interesting leads for drug development against numerous diseases [1,2,3]. Among many other examples, the class of germacranolides has attracted a great deal of attention in recent years. Parthenolide, a germacranolide isolated from *Tanacetum parthenium*, exhibited promising antitumor efficacy [3,4]. Germacranolides are one class of the main sesquiterpene lactones, reported with broad bioactivities, including cytotoxicity, anti-inflammation, and antimalarial action [4,5,6,7]. In the past five years, germacranolides have been reported in more than 250 publications [4]. These germacranolides contain a plethora of stereogenic centers and a multitude of oxygenated functionalities, creating the problem of the assignment of absolute configuration.

In our ongoing search for new/novel and bioactive products from the medicinal plants in China, *Carpesium divaricatum* Sieb.et Zucc, belonging to the family Compositae, were found to be rich in highly oxygenated germacranolides [6,7,8,9,10]. Our previous study led to the distinction between four subtypes of these germacranolides [11,12,13,14]. A further investigation of *C. divaricatum* was conducted, resulting in the isolation of eight new 11-methoxymethylgermacranolides (**1**–**8**). Notably, compounds **1**–**8** represent a new subtype V (named 3-oxo-11-methoxymethylgermacranolide), possessing a 6,7-*γ*-lactone ring and the 3-ketone group. Subtypes IV (the basic structure of cardivarolides) and V have similar skeletons except for the presence of a methoxymethyl group instead of the *Δ*^11,13^ exocyclic methylene group in the five-membered ring [12,13] (Figure 1). In this paper, the isolation, structural elucidation, absolute configuration, and bioactive evaluation of these compounds are presented.

## 2. Results and Discussion

### 2.1. Structural Elucidation of the Isolated Compounds

Compound **1** (Figure 2) was obtained as white needles. The molecular formula was assigned as C_25_H_38_O_10_ on the basis of the positive-ion HRESIMS peak at *m/z* 521.2366 [M + Na]^+^, together with its ^1^H and ^13^C NMR data (Table 1 and Table 2). Its IR spectrum showed hydroxyl (3441 cm^−1^) and carbonyl (1758 and 1717 cm^−1^) absorptions. The ^1^H NMR data indicated the presence of a methoxy group at *δ*_H_ 3.35 (3H, s, H-16); four oxygenated methine groups *δ*_H_ 5.44 (1H, d, *J* = 9.6 Hz, H-5), 4.52 (1H, dd, *J* = 9.6, 9.0 Hz, H-6), 4.53 (1H, o, H-8), and 4.94 (1H, d, *J* = 10.2 Hz, H-9); an isobutyryloxy group at *δ*_H_ 2.65 (1H, m, H-2′), 1.20 (3H, d, *J* = 7.2 Hz, H_3_-3′), and 1.19 (3H, d, *J* = 7.2 Hz, H_3_-4′); an angeloyloxy group at *δ*_H_ 6.12 (1H, q, *J* = 7.2 Hz, H-3′′), 1.93 (3H, s, H_3_-4′′), and 1.98 (3H, br d, *J* = 7.2 Hz, H_3_-5′′) and two additional methyl groups at *δ*_H_ 0.88 (3H, d, *J* = 6.6 Hz, H_3_-14) and 1.18 (3H, s, H_3_-15). The ^13^C NMR spectrum showed the presence of 25 carbon signals, in which the characteristic carbon signals at *δ*_C_ 217.4 (C-3), 176.3 (C-1′), 168.0 (C-1′′), 80.4 (C-4), 79.0 (C-5), 80.3 (C-6), 67.3 (C-8), and 79.3 (C-9) were readily assigned. These data and the carbon signals at *δ*_C_ 177.0 (C-12) and 69.7 (C-13) indicated that **1** is an 11-methoxymethylgermacranolide with an isobutyryloxy group and the angeloyloxy group [15]. The locations of the two substituted groups at C-5 and C-9 were based on the HMBC correlations of H-5 (*δ*_H_ 5.44)/C-1′ (*δ*_C_ 176.3) and H-9 (*δ*_H_ 4.94)/C-1′′ (*δ*_C_ 168.0) (Figure 3). These observations were further confirmed by analyses of relevant ^1^H-^1^H COSY and HSQC data (Figure 3). On the basis of these data, the planar structure of **1** was established.

The relative configuration of **1** was determined by analysis of NOESY data (Figure 3). The NOE associations of H_3_-15/H-5, H-5/H-7, H-7/H-9, and H-9/H-10 revealed that these protons were *α*-oriented. The NOESY correlations from H-6 to H-8, from H-8 to H-11, and from H-6 to H-11 suggested the *β*-orientations of H-6, H-8, and H-11. These orientations were confirmed by Cu K*α* X-ray crystallographic analysis (Figure 4). Thus, the structure of compound **1** was defined as (4*R*, 5*R*, 6*S*, 7*R*, 8*R*, 9*R*, 10*R*, 11*R*)-5-isobutyryloxy-4,8-dihydroxy-11-methoxymethyl-9-angeloyloxy-3-oxogermacran-6,12-olide, named 11-methoxymethylcardivarolide H.

The molecular formula of compound **2** was assigned as C_25_H_40_O_10_ by positive-ion HRESIMS at *m/z* 523.2526 [M + Na]^+^. Furthermore, the ^1^H and ^13^C NMR data implied that the structure of **2** was similar to that of **1**, except that the angeloyloxy group at C-9 in **1** was replaced by a 3-methylbutyryloxy group in **2**, which was confirmed by the HMBC correlations of H_3_-4′′ and H_3_-5′′/C-2′′. The key NOE correlations of H_3_-15/H-5, H-5/H-7, H-7/H-9, H-8/H-6, H-8/H-11, and H-6/H-11 indicated that **2** had the same relative configuration as **1**. The ECD spectrum of **2** showed a positive Cotton effect at near 225 nm and a negative Cotton effect at near 306 nm, which closely resembled those of **1** (Appendix A). Based on biosynthetic considerations [4,13], similar ROESY and ECD data of **2** and **1** assigned the absolute configuration of **2** as 4*R*, 5*R*, 6*S*, 7*R*, 8*R*, 9*R*, 10*R*, and 11*R*. Thus, the structure of compound **2** was elucidated as shown in Figure 2, named 11-methoxymethylcardivarolide I.

Compounds **3**–**4** had molecular formulas of C_24_H_38_O_10_ and C_24_H_36_O_10_ according to their HRESIMS ions at *m/z* 509.2369 [M + Na]^+^ and *m/z* 507.2202 [M + Na]^+^, respectively. The NMR data of **3**–**4** were comparable with those of **1**, except for the presence of an isobutyryloxy group in **3** and a 2-methacryloyloxy group in **4** instead of the angeloyloxy group at C-9 in **1**, respectively. These observations were confirmed by analyses of relevant ^1^H−^1^H COSY, HSQC, and HMBC data. The relative configurations of **3**–**4** were determined to be the same as that of **1** by comparison of ROESY data for relevant protons. Based on biosynthetic considerations [4,13], similar ECD data of **3**–**4** and **1** revealed the same absolute configurations of **3**–**4** as that of **1**. Thus, the structures of **3**–**4** were depicted as shown and were named 11-methoxymethylincaspitolide D and 11-methoxymethylcardivarolide J, respectively.

Compounds **5**–**6** shared the same molecular formula C_26_H_42_O_10_, established from their HRESIMS ions at *m/z* 537.2686 [M + Na]^+^ and *m/z* 537.2688 [M + Na]^+^. The ^1^H and ^13^C NMR data of **5**–**6** showed great similarity with those of **1**, except for the ester residues at C-5 and C-9. An isobutyryloxy group at C-5 and an angeloyloxy group at C-9 in **1** were placed by two 3-methylbutyryloxy groups of **5** and a 2-methylbutyryloxy group and a 3-methylbutyryloxy group of **6**, respectively. These observations were confirmed by analyses of relevant ^1^H−^1^H COSY, HSQC, and HMBC data. Similarly, their relative configurations were determined as the same as that of **1** by comparison of the ROESY data. The absolute configuration of **5** was established by using quantum chemical electronic circular dichroism (ECD) calculations. Due to the huge amounts of conformations from its numerous single bonds, a simplified structure named **5Ja** (Appendix A), in which two acetyl groups instead of the 3-methylbutyryloxy moieties, was used for ECD calculations [16]. The calculated ECD spectrum (Figure 5) of (4*R*, 5*R*, 6*S*, 7*R*, 8*R*, 9*R*, 10*R*, 11*R*)-**5Ja** agreed well with the experimental spectrum and confirmed the (4*R*, 5*R*, 6*S*, 7*R*, 8*R*, 9*R*, 10*R*, 11*R*) absolute configuration. Based on biosynthetic considerations [4,13], similar ECD data of **6** and **5** revealed the same absolute configuration of **6** as that of **5**. Thus, the structures of compounds **5**–**6** were established, as shown in Figure 2, named 11-methoxymethylcardivarolide K and 11-methoxymethylcardivarolide L, respectively.

The HRESIMS data of compounds **7**–**8** suggested the molecular formulas of C_26_H_40_O_10_ and C_26_H_38_O_10_, respectively. The NMR data of **7** were similar to those of **5**, except that an angeloyloxy group appeared in **7** instead of a 3-methylbutyryloxy group at C-5 in **5**. For the same reason, the NMR data implied the presence of an angeloyloxy group at C-9 rather than a 3-methylbutyryloxy group in **8** compared to **7**. The ^1^H−^1^H COSY, HSQC, and HMBC spectra of **7**–**8** confirmed these observations, leading to the assignment of their planar structures. The relative configurations of **7**–**8** were deduced to be the same as **1** on the basis of similar ROESY data. Considering similar ECD data of **7**–**8** and **1** resulted in the conclusion of their same absolute configurations. Due to the fact that there are some differences in the ECD spectra of **1** and **7**, the absolute configuration of **7** was further confirmed by ECD calculations. Similarly, a simplified structure named **7Ja** (Appendix A), in which an acetyl group instead of a 3-methylbutyryloxy moiety was used for ECD calculations [16]. It was clear that the calculated ECD spectrum of (4*R*, 5*R*, 6*S*, 7*R*, 8*R*, 9*R*, 10*R*, 11*R*)-**7Ja** was matched very well with the experimental ECD spectrum of **7** (Figure 6). Thus, the structures of **7**–**8** were elucidated and were named 11-methoxymethylcardivarolide G and 11-methoxymethylcardivarolide F.

### 2.2. Cytotoxic Activity

All compounds were evaluated for their cytotoxic activity against human hepatocellular cancer (Hep G2), breast cancer (MCF-7), and lung cancer (A549) cell lines. Only new compound **4** exhibited weak cytotoxicity against MCF-7 (IC_50_ value of 37.32 μM), compared with the positive control *cis*-platin (IC_50_ value of 22.80 μM) (Table 3).

### 2.3. Analysis of the Macrophages Culture Supernatants PGE2 Levels

Numerous studies have suggested the biologically pivotal roles of PGE2 in cancer, inflammation, and pain [17,18,19,20,21]. Due to the insufficient amount of isolates, only compounds **2**, **3,** and **8** were tested for the effects on PGE2 production in the supernatant of LPS-induced RAW 264.7 cells by a highly sensitive ELISA in this study. LPS stimulation resulted in a marked increase in PGE2 in the macrophage culture supernatants or the mice sera. Among the three compounds, pretreatment with **8** could potently decrease PGE2 contents, even lower than the normal level (Figure 7).

## 3. Materials and Methods

### 3.1. General Experimental Procedures

Optical rotations were measured on a Perkin-Elmer 241 polarimeter (Perkin-Elmer, Waltham, MA, USA), and UV spectra were recorded on Shimadzu UV-2501 PC (Shimadzu, Kyoto, Japan). IR data were recorded using a Shimadzu FTIR-8400S spectrophotometer (Shimadzu, Kyoto, Japan). ^1^H and ^13^C-NMR data were acquired with Bruker 600 and Bruker 500 instruments (Bruker, Rheinstetten, Germany) using the solvent signals as references. High-resolution electrospray ionization mass spectroscopy (HRESIMS) data were acquired using a Q-TOF analyzer in the SYNAPT HDMS system (Waters, Milford, MA, USA). ECD spectra were recorded on a JASCO J-815 Spectropolarimeter (Jasco, Tokyo, Japan). X-ray diffraction data were collected on the Agilent GEMINITME instrument (CrysAlisPro software, Version 1.171.35.11; Agilent, Santa Clara, CA, USA). High-performance liquid chromatography (HPLC) was performed using the Waters 2535 system (Waters, Milford, MA, USA) with the following components: preparative column, a Daisogel-C18-100A (10 μm, 30 × 250 mm, ChuangXinTongHeng Sci.&Tech., Beijing, China), a YMC-Pack ODS-A column (5 μm, 10 × 250 mm, YMC, Kyoto, Japan), and a detector (Waters 2489 UV). Sephadex LH-20 (40–70 µm, Pharmacia Biotech AB, Uppsala, Sweden), silica gel (60–100, 100–200, and 200–300 mesh), and silica gel GF254 sheets (0.20–0.25 mm) (Qingdao Marine Chemical Plant, Qingdao, China) were used for column chromatography and thin-layer chromatography (TLC), respectively. TLC spots were visualized under UV light and by dipping into 5% H_2_SO_4_ in EtOH followed by heating.

### 3.2. Plant Material

The whole plant of *C. divaricatum* was collected from EnShi, Hubei province of China (GPS coordinates: 109°29′11.586″N, 30°18′1.945″E) in August of 2013. They were identified by Prof. Ben-Gang Zhang of the Institute of Medicinal Plant Development. A voucher specimen (No. 20130828) was deposited in the National Compound Library of Traditional Chinese Medicines, Institute of Medicinal Plant Development, Chinese Academy of Medical Sciences & Peking Union Medical College (CAMS & PUMC), China.

### 3.3. Extraction and Isolation

The air-dried plants (9 kg) were extracted three times (7 days each time) with EtOH–H_2_O (95:5) at room temperature. The combined extract was concentrated under reduced pressure to furnish a dark brown residue (570 g), which was suspended in H_2_O and partitioned in turn with petroleum ether (bp 60–90 °C), ethyl acetate (EtOAc), and n-butyl alcohol (n-BuOH). The EtOAc extract (207 g) was separated chromatographically on silica gel column (60–100 mesh, 16 × 20 cm) with a gradient mixture of CH_2_Cl_2_–MeOH (100:1, 60:1, 30:1, 15:1, and 6:1) as eluent. Five fractions were collected according to TLC analysis. Fraction A (CH_2_Cl_2_–CH_3_OH, 100:1, 140 g) was separated by silica gel column chromatography (CC) (100–200 mesh, 16 × 20 cm) with petroleum ether–acetone (50:1, 25:1, 20:1, 15:1, 12:1, 10:1, 7:1, 5:1, 3:1, and 1:1) as eluent to give fractions A_1_–A_11_. Fraction A_10_ (petroleum ether–acetone, 3:1, 40 g) was separated by Sephadex LH-20 CC (5 × 200 cm, CH_3_OH) to give Fr.A_10_S_1_–Fr.A_10_S_3_. Fraction A_10_S_2_ (20 g) was then subjected to MCI gel CC (6 × 50 cm) with a gradient mixture of CH_3_OH–H_2_O (60:40, 80:20, and 100:0, 4000 mL each) to give three fractions (Fr.A_10_S_2_M_1_–Fr.A_10_S_2_M_3_).

Fraction A_10_S_2_M_2_ (13 g) was further separated chromatographically on silica gel column (200–300 mesh, 5 × 50 cm) with a gradient mixture of CH_2_Cl_2_–MeOH (150:1, 100:1, 50:1, and 20:1) as eluent, and a total of 86 fractions (Fr.A_10_S_2_M_2_-1–86, 200 mL each) were collected. Fraction A_10_S_2_M_2_-20–24 (2 g) were separated by preparative HPLC (20 mL/min, 65% CH_3_OH in H_2_O) and semipreparative HPLC (2 mL/min, 60% CH_3_OH in H_2_O for 10 min, and followed by 60–90% CH_3_OH in H_2_O for 25 min; 2 mL/min, 40–85% CH_3_CN in H_2_O for 40 min) to yield **8** (10 mg). Fraction A_10_S_2_M_2_-34–50 (1.5 g) were separated by preparative HPLC (20 mL/min, 70% CH_3_OH in H_2_O) and semipreparative HPLC (2 mL/min, 52–75% CH_3_OH in H_2_O for 25 min, and followed by 75–95% CH_3_OH in H_2_O for 10 min; 2 mL/min, 40–80% CH_3_CN in H_2_O for 40 min) to yield **2** (10 mg), **3** (10 mg), and **4** (6 mg). Fraction A_10_S_2_M_2_-74–79 (140 mg) were purified using semipreparative HPLC (2 mL/min, 60–80% CH_3_OH in H_2_O for 25 min and followed by 80–90% CH_3_OH in H_2_O for 20 min; 2 mL/min, 30–70% CH_3_CN in H_2_O for 40 min) and to yield **7** (5 mg).

Fraction A_9_ (petroleum ether–acetone, 5:1, 30 g) was separated by Sephadex LH–20 CC (5 × 200 cm, CH_3_OH) to give Fr.A_9_S_1_–Fr.A_9_S_3_. Fraction A_9_S_2_ (20 g) was then subjected to MCI gel CC (6 × 50 cm) with a gradient mixture of CH_3_OH–H_2_O (60:40, 80:20, and 100:0, 4000 mL each) to give three fractions (Fr.A_9_S_2_M_1_–Fr.A_9_S_2_M_3_). Fraction A9S2M2 (10 g) was further separated chromatographically on silica gel column (100–200 mesh, 5 × 50 cm) with a gradient mixture of petroleum ether–acetone (10:1, 7:1, 5:1, 3.5:1, 2:1, and 1:1) as eluent, and a total of 200 fractions (Fr.A_9_S_2_M_2_-1–200, 50 mL each) were collected. Fraction A_9_S_2_M_2_-113–123 (1 g) were separated by preparative HPLC (20 mL/min, 65% CH_3_OH in H_2_O) and semipreparative HPLC (2 mL/min, 68% CH_3_OH in H_2_O for 50 min; 2 mL/min, 40–80% CH_3_CN in H_2_O for 40 min) to yield **5** (4.7 mg), **6** (12.5 mg), and **1** (5.5 mg).

### 3.4. Spectral Data

11-methoxymethylcardivarolide H (**1**)

White needles (CH_3_OH); ${\left[\alpha \right]}_{\;D}^{20}$ −68.8 (*c* 0.125, CH_3_OH); UV (CH_3_OH) *λ*_max_ (log*ε*) 216 (3.84) nm; IR (neat) *ν*_max_ 3441, 1758, 1717, 1633 cm^−1^; ECD (CH_3_OH) 305 (Δ*ε* −0.036) nm; HRESIMS (pos.) *m/z* 521.2366 [M + Na]^+^ (calcd. for C_25_H_38_O_10_Na, 521.2363); ^1^H NMR data, see Table 1; ^13^C NMR data, see Table 2.

11-methoxymethylcardivarolide I (**2**)

White needles (CH_3_OH); ${\left[\alpha \right]}_{\;D}^{20}$ −84.8 (*c* 0.165, CH_3_OH); UV (MeOH) *λ*_max_ (log*ε*) 206 (2.94) nm; IR (KBr) *ν*_max_ 3462, 1744, 1718 cm^−1^; ECD (CH_3_OH) 229 (Δ*ε* +0.005), 307 (Δ*ε* −0.035) nm; HRESIMS (pos.) *m/z* 523.2526 [M + Na]^+^ (calcd. for C_25_H_40_O_10_Na, 523.2519); ^1^H NMR data, see Table 1; ^13^C NMR data, see Table 2.

11-methoxymethylincaspitolide D (**3**)

White needles (CH_3_OH); ${\left[\alpha \right]}_{\;D}^{20}$ −86.4 (c 0.110, CH_3_OH); UV (CH_3_OH) *λ*_max_ (log*ε*) 206 (3.52) nm; IR (neat) *ν*_max_ 3437, 1750, 1729, 1652 cm^−1^; ECD (CH_3_OH) 308 (Δ*ε* −0.012) nm; HRESIMS (pos.) *m/z* 509.2369 [M + Na]^+^ (calcd. for C_24_H_38_O_10_Na, 509.2363); ^1^H NMR data, see Table 1; ^13^C NMR data, see Table 2.

11-methoxymethylcardivarolide J (**4**)

White needles (CH_3_OH); ${\left[\alpha \right]}_{\;D}^{20}$ −40.0 (c 0.140, CH_3_OH); UV (CH_3_OH) *λ*_max_ (log*ε*) 205 (3.60) nm; IR (neat) *ν*_max_ 3474, 1764, 1719 cm^−1^; ECD (CH_3_OH) 307 (Δ*ε* −0.020) nm; HRESIMS (pos.) *m/z* 507.2202 [M + Na]^+^ (calcd. for C_24_H_36_O_10_Na, 507.2202); ^1^H NMR data, see Table 1; ^13^C NMR data, see Table 2.

11-methoxymethylcardivarolide K (**5**)

White needles (CH_3_OH); ${\left[\alpha \right]}_{\;D}^{20}$ −82.2 (c 0.135, CH_3_OH); UV (CH_3_OH) *λ*_max_ (log*ε*) 197 (3.55) nm; IR (neat) *ν*_max_ 3452, 1740, 1715, 1632 cm^−1^; ECD (CH_3_OH) 238 (Δ*ε* +0.004), 306 (Δ*ε* −0.046) nm; HRESIMS (pos.) *m/z* 537.2686 [M + Na]^+^ (calcd. for C_26_H_42_O_10_Na, 537.2676); ^1^H NMR data, see Table 1; ^13^C NMR data, see Table 2.

11-methoxymethylcardivarolide L (**6**)

White needles (CH_3_OH); ${\left[\alpha \right]}_{\;D}^{20}$ −76.8 (c 0.125, CH_3_OH); UV (CH_3_OH) *λ*_max_ (log*ε*) 201 (2.97) nm; IR (neat) *ν*_max_ 3457, 1749, 1736, 1706 cm^−1^; ECD (CH_3_OH) 228 (Δ*ε* +0.008), 306 (Δ*ε* −0.050) nm; HRESIMS (pos.) *m/z* 537.2688 [M + Na]^+^ (calcd. for C_26_H_42_O_10_Na, 537.2676); ^1^H NMR data, see Table 1; ^13^C NMR data, see Table 2.

11-methoxyldivarolide G (**7**)

White needles (CH_3_OH); ${\left[\alpha \right]}_{\;D}^{20}$ −48.0 (c 0.150, CH_3_OH); UV (CH_3_OH) *λ*_max_ (log*ε*): 216 (3.66) nm, IR (neat) *ν*_max_ 3456, 1760, 1733, 1709 cm^−1^; ECD (CH_3_OH) 222 (Δ*ε* +0.023), 307 (Δ*ε* −0.037) nm; HRESIMS (pos.) *m/z* 535.2511 [M + Na]^+^ (calcd. for C_26_H_40_O_10_Na, 535.2519); ^1^H NMR data, see Table 1; ^13^C NMR data, see Table 2.

11-methoxyldivarolide F (**8**)

White needles (CH_3_OH); ${\left[\alpha \right]}_{\;D}^{20}$ −36.9 (c 0.160, CH_3_OH); UV (CH_3_OH) *λ*max (log*ε*): 205 (3.59) nm; IR (neat) *ν*_max_: 3463, 1745, 1716 cm^−1^; ECD (CH_3_OH) 239 (Δ*ε*; −0.022), 306 (Δ*ε* −0.047) nm; HRESIMS (pos.) *m/z* 533.2374 [M + Na]^+^ (calcd. for C_26_H_38_O_10_Na, 533.2363); ^1^H NMR data, see Table 1; ^13^C NMR data, see Table 2.

### 3.5. X-ray Crystal Structure Analysis

X-ray diffraction data were collected on the Agilent GEMINITME instrument (CrysAlisPro software, Version 1.171.35.11), with enhanced Cu K*α* radiation (*λ* = 1.54184 Å). The structure was solved by direct methods and refined by full-matrix least-squares techniques (SHELXL-97). All non-hydrogen atoms were refined with anisotropic thermal parameters. Hydrogen atoms were located by geometrical calculations and from positions in the electron density maps. Crystallographic data (excluding structure factors) for **1** in this paper have been deposited with the Cambridge Crystallographic Data Centre (Deposition Number: CCDC 1846500). Copies of the data can be obtained, free of charge, on application to CCDC, 12 Union Road, Cambridge CB2 1EZ, UK (fax: +44-12-23336033 or e-mail: deposit@ccdc.cam.ac.uk).

A colorless monoclinic crystal (0.25 × 0.22 × 0.13 mm) of **1** was grown from CH_3_OH. Crystal data: C_25_H_38_O_10_, M = 498.55, T = 110.7 K, monoclinic, space group P2_1_, a = 9.0365 (3) Å, b = 11.1281 (3) Å, c = 13.1083 (4) Å, *α* = 90.00°, *β* = 97.237 (3), γ = 90.00°, V = 1307.66 (7) Å^3^, Z = 2, ρ = 1.266 mg/mm^3^, μ (Cu K*α*) = 0.813 mm^−1^, measured reflections = 8728, unique reflections = 4912 (R_int_ = 0.0283), largest difference peak/hole = 0.210/−0.177 eÅ^−3^, and flack parameter = −0.03 (12). The final R indexes [I > 2σ (I)] were R_1_ = 0.0327, and wR_2_ = 0.0818. The final R indexes (all data) were R_1_ = 0.0341, and wR_2_ = 0.0831. The goodness of fit on F^2^ was 1.035.

### 3.6. Biological Activity Assays

Cell cultures: Human HepG2, MCF-7, and A549 cell lines from the Cancer Institute and Hospital of Chinese Academy of Medical Sciences and RAW264.7 cells were purchased from the American Type Culture Collection (Manassas, VA, USA), respectively. They were cultured in Dulbecco’s modified Eagle’ s medium (DMEM, Gibco, Gaithersburg, MD, USA) supplemented with 10% (*v/v*) fetal calf serum (Gibco, USA), penicillin G (Macgene, Beijing, China) 100 units mL^−1^, and streptomycin (Macgene, China), 100 μg mL^−1^, at 37 °C under 5% CO_2_.

Cell viability assay: The assay was run in triplicate. In a 96-well plate, each well was plated with 2 × 10^4^ cells. After cell attachment overnight, the medium was removed, and each well was treated with 100 μL of medium containing 0.1% DMSO or different concentrations of the test compounds and the positive control cis-platin. The plate was incubated for 4 days at 37 °C in a humidified, 5% CO_2_ atmosphere. Cytotoxicity was determined using a modified 3-(4,5-dimethylthiazol-2-yl)-2,5-diphenyltetrazolium bromide (MTT) colorimetric assay [22]. After addition of 10 μL MTT solution (5 mg/mL), cells were incubated at 37 °C for 4 h. After adding 150 μL DMSO, cells were shaken to mix thoroughly. The absorbance of each well was measured at 540 nm in a multiscan photometer. The IC_50_ values were calculated by Origin software.

The culture supernatant assay: For the culture supernatant assay, RAW 264.7 cells were pretreated with the tested compounds (10 µM) for 2 h and then stimulated with LPS (10 μg/L) for 24 h. PGE2 concentrations in the culture supernatants were simultaneously assayed by hscELISA. At least 10- and 50-fold dilutions are needed for the culture supernatant tests.

Statistical analysis: Values were expressed as mean ± SD. Statistical analyses were performed using the Student’s *t*-test. Differences were considered significant when associated with a probability of 5 % or less (*p* ≤ 0.05).

## 4. Conclusions

In conclusion, eight new compounds (1–8) representing a new subtype (subtype V, named 3-oxo-11-methoxymethylgermacranolide) of germacranolides, were isolated from the whole plant of *C. divaricatum*. Notably, a pair of isomers (5/6) was obtained from the same plant. The absolute configuration of compound 1 was unambiguously established by X-ray diffraction. The other compounds with the same skeleton were determined by comparison of NOESY and ECD data with those of 1. Structurally, all compounds contained a 5-membered *γ*-lactone ring with the methoxymethyl group fused to a circular 10-membered carbocycle. Based on the common structural features, these germacranolide analogs are different as far as substituents are concerned. Compound 4 showed weak cytotoxicity against a human tumor cell line. Compound 8 could potently decrease PGE2 productions in LPS-induced RAW 264.7 cells. These findings are an important addition to the present knowledge on the structurally diverse and biologically important germacranolide family.

## Figures and Tables

**Figure 1 molecules-27-05991-f001:**
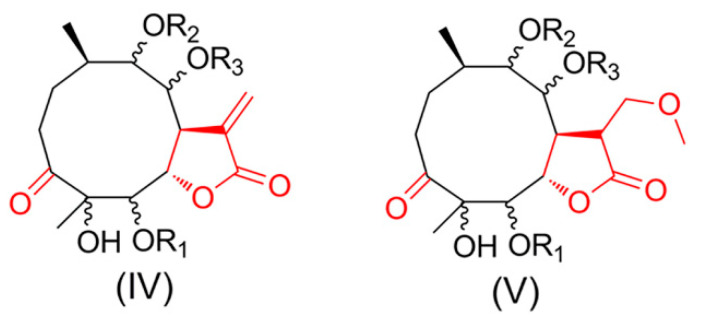
Subtypes IV and V of germacranolides. The basic structure of cardivarolides is same as subtypes IV.

**Figure 2 molecules-27-05991-f002:**
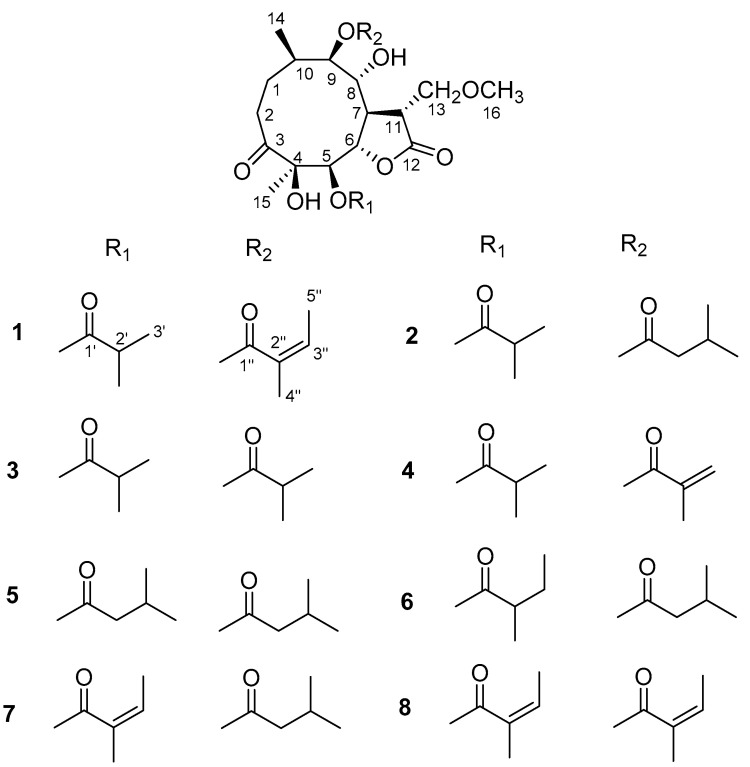
Structures of compounds **1**–**8**.

**Figure 3 molecules-27-05991-f003:**
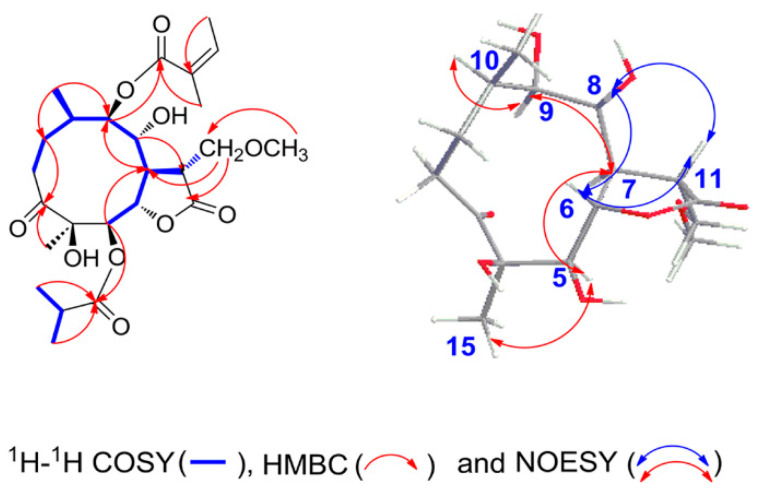
Key ^1^H−^1^H COSY, HMBC, and NOESY correlations of **1**.

**Figure 4 molecules-27-05991-f004:**
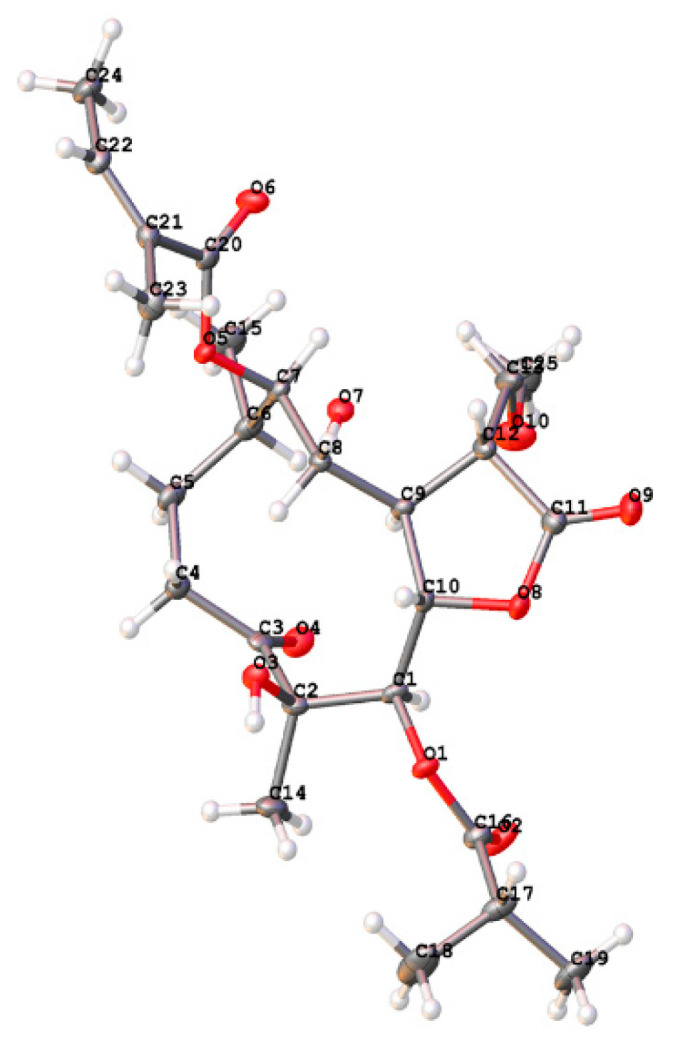
X-ray ORTEP drawing of **1**.

**Figure 5 molecules-27-05991-f005:**
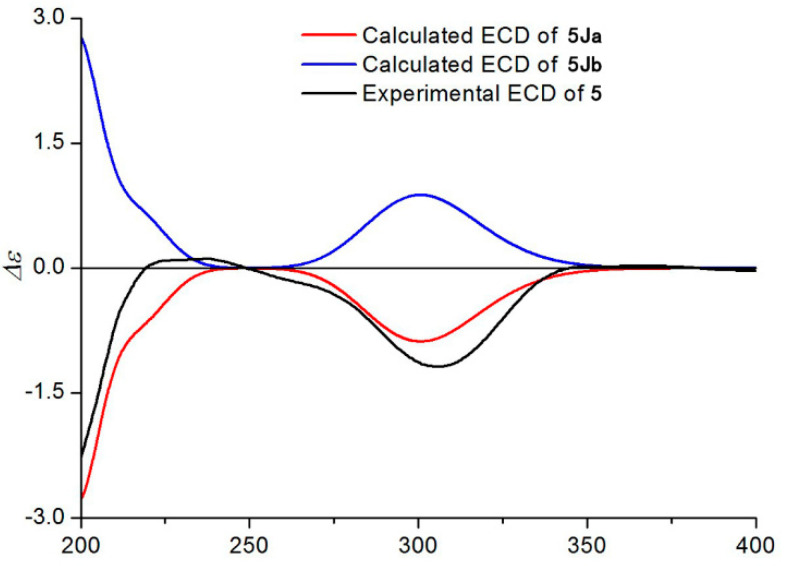
Experimental ECD spectrum of **5** and calculated ECD spectra of **5Ja** and **5Jb**.

**Figure 6 molecules-27-05991-f006:**
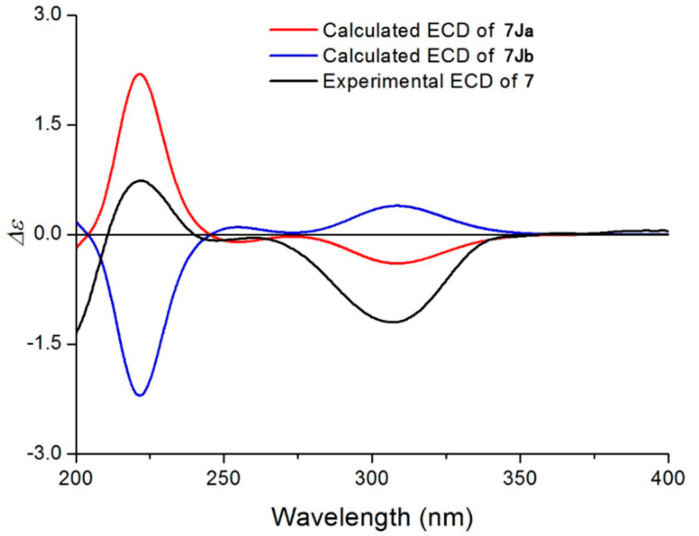
Experimental ECD spectrum of **7** and calculated ECD spectra of **7Ja** and **7Jb**.

**Figure 7 molecules-27-05991-f007:**
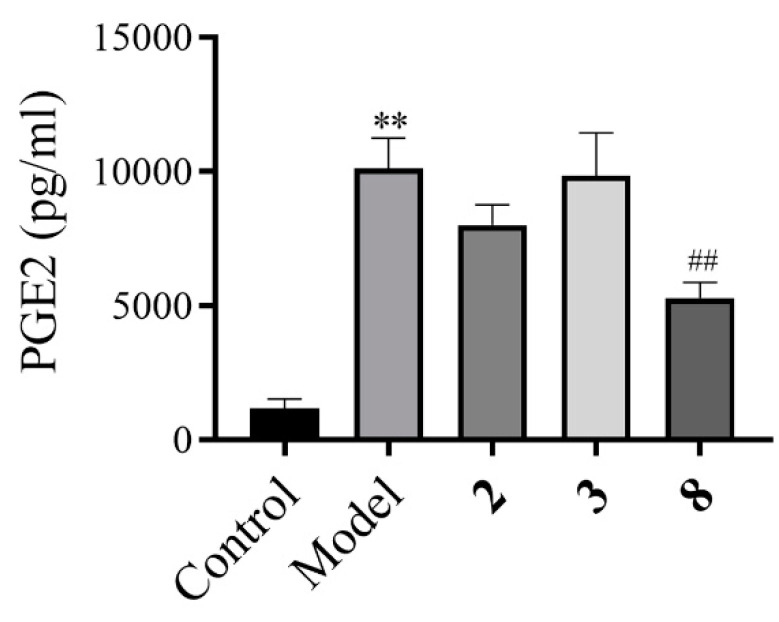
Analysis of the macrophages culture supernatants prostaglandin E2 levels by a highly sensitive ELISA, when treated with compounds **2**, **3**, and **8**. ** *p* < 0.01 vs. control group, ^##^
*p* < 0.01 vs. model group.

**Table 1 molecules-27-05991-t001:** ^1^H NMR spectroscopic data for compounds **1**–**8** (*δ* in ppm, *J* in Hz).

No.	1 ^a^	2 ^a^	3 ^a^	4 ^a^	5 ^a^	6 ^a^	7 ^b^	8 ^a^
1	1.86 m, 1.55 m	1.81 m, 1.54 m	1.82 m, 1.57 m	1.85 m, 1.57 m	1.82 m, 1.54 m	1.82 m, 1.51 o	1.86 m, 1.57 m	1.87 m, 1.55 m
2	3.89 dd (12.6, 3.6), 2.00 o	3.87 dd (12.6, 3.6), 2.09 o	3.89 br d (12.0), 2.10 o	3.91 br d (12.6), 2.00 m	3.87 dd (12.6, 3.6), 2.13 m	3.87 dd (12.6, 3.6), 2.09 o	3.92 dd (12.0, 2.0), 2.15 m	3.92 dd (12.6, 3.6), 1.97 o
5	5.44 d (9.6)	5.43 dd (9.0, 1.8)	5.43 br d (9.6)	5.43 dd (9.0, 1.2)	5.44 d (9.6)	5.44 dd (9.6, 1.8)	5.56 dd (9.5, 1.5)	5.54 dd (9.0, 1.8)
6	4.52 dd (9.6, 9.0)	4.52 dd (9.0, 9.0)	4.50 dd (9.6, 9.0)	4.52 dd (9.0, 9.0)	4.50 dd (9.6, 9.0)	4.54 dd (9.6, 9.0)	4.60 dd (9.0, 8.5)	4.58 dd (9.0, 9.0)
7	2.60 dd (9.0, 9.0)	2.56 dd (9.0, 9.0)	2.57 dd (9.0, 9.0)	2.59 dd (9.0, 9.0)	2.57 dd (9.0, 8.4)	2.57 dd (9.0, 8.4)	2.62 dd (9.0, 8.5)	2.62 dd (9.0, 9.0)
8	4.53 o	4.47 br d (10.2)	4.55 dd (8.4, 8.4)	4.54 br d (10.2)	4.47 br d (10.8)	4.48 br d (10.2)	4.52 d (10.5)	4.55 br d (10.2)
9	4.94 d (10.2)	4.85 o	4.82 o	4.95 d (10.2)	4.85 o	4.85 o	4.87 o	4.95 d (10.2)
10	2.15 m	2.12 m	2.12 m	2.15 m	2.10 o	2.12 m	2.16 m	2.16 m
11	3.31 ddd (9.0, 6.0, 3.6)	3.26 ddd (10.8, 7.8, 4.2)	3.25 ddd (9.0, 5.4, 4.2)	3.29 ddd (10.2, 7.8, 4.2)	3.26ddd (9.0, 7.8, 4.2)	3.25 ddd (9.0, 7.8, 3.6)	3.32 o	3.33 ddd (9.0, 7.8, 3.0)
13a	3.67 dd (9.6, 4.2)	3.65 dd (9.6, 4.2)	3.66 dd (9.6, 4.2)	3.66 dd (9.6, 4.2)	3.65 dd (10.2, 4.2)	3.66 dd (10.2, 4.2)	3.70 dd (9.5, 4.5)	3.70 dd (10.2, 4.2)
13b	3.43 dd (9.6, 4.2)	3.40 dd (9.6, 4.2)	3.40 dd (9.6, 4.2)	3.42 dd (9.6, 4.2)	3.41 dd (10.2, 4.2)	3.40 dd (10.2, 4.2)	3.43 dd (9.5, 4.0)	3.45 dd (10.2, 4.2)
14	0.88 d (6.6)	0.87 d (7.2)	0.86 d (7.2)	0.87 d (7.2)	0.87 d (6.6)	0.87 d (7.2)	0.90 d (7.0)	0.89 d (6.6)
15	1.18 s	1.17 s	1.18 s	1.18 s	1.18 s	1.18 s	1.22 s	1.19 s
16	3.35 s	3.34 s	3.34 s	3.35 s	3.34 s	3.34 s	3.37 s	3.36 s
2′	2.65 m	2.64 m	2.64 o	2.65 m	2.31 d (6.6), 2.23 d (6.6)	2.47 m		
3′	1.20 d (7.2)	1.20 d (7.2)	1.19 d (6.6)	1.20 d (7.2)	2.10 o	1.73 m, 1.51 o	6.17 qq (7.0, 1.5)	6.13 o
4′	1.19 d (7.2)	1.19 d (7.2)	1.18 d (6.6)	1.19 d (7.2)	0.97 d (6.6)	1.17 d (7.2)	1.95 qq (1.5, 1.5)	1.92 s
5′					0.97 d (6.6)	0.94 t (7.2)	0.97 dq (7.0, 1.5)	1.97 d (9.0)
2′′		2.28 d (7.2), 2.27 d (6.6)	2.64 o		2.27 o, 2.27 o	2.28 d (7.2), 2.27 d (7.2)	2.31 d (7.0), 2.30 d (7.0)	
3′′	6.12 q (7.2)	2.09 o	1.20 d (7.2)	5.63 dq (3.6, 1.8), 6.12 dq (3.6, 1.8)	2.10 o	2.09 o	2.13 m	6.13 o
4′′	1.93 s	0.98 d (6.6)	1.16 d (7.2)	1.96 br s	0.97 d (6.6)	0.97 d (7.2)	1.01 d (6.5)	1.92 s
5′′	1.98 br d (7.2)	0.97 d (6.6)			0.97 d (6.6)	0.97 d (6.6)	1.00 d (6.5)	1.97 d (9.0)

^a^ Measured at 600 MHz in methanol-*d*_4_; ^b^ Measured at 500 MHz in methanol-*d*_4_; o: Overlapped with other signals.

**Table 2 molecules-27-05991-t002:** ^13^C NMR spectroscopic data for compounds **1**–**8** (*δ* in ppm).

No.	1 ^a^	2 ^a^	3 ^a^	4 ^a^	5 ^a^	6 ^a^	7 ^b^	8 ^a^
1	25.3	25.3	25.3	25.3	25.4	25.3	25.4	25.6
2	31.6	31.6	31.7	31.7	31.9	31.5	31.6	31.6
3	217.4	217.5	217.5	217.5	217.4	217.4	217.7	217.7
4	80.4	80.4	80.4	80.4	80.3	80.3	80.4	80.4
5	79.0	79.0	79.0	79.0	79.0	78.9	78.9	78.9
6	80.3	80.3	80.3	80.3	80.3	80.3	80.4	80.4
7	39.3	39.2	39.2	39.2	39.2	39.2	39.2	39.3
8	67.3	67.1	67.1	67.1	67.1	67.1	67.1	67.3
9	79.3	79.5	79.3	79.3	79.5	79.5	79.3	79.3
10	29.4	29.2	29.4	29.4	29.3	29.3	29.4	29.4
11	40.2	40.2	40.1	40.1	40.2	40.2	40.2	40.2
12	177.0	177.0	177.0	177.0	176.9	176.8	177.0	177.1
13	69.7	69.7	69.7	69.7	69.8	69.8	69.7	69.7
14	20.0	20.0	19.9	19.9	20.0	20.0	20.0	20.0
15	23.5	23.6	23.6	23.6	23.7	23.6	23.6	23.6
16	58.0	57.9	57.9	57.9	58.0	58.0	58.0	58.0
1′	176.3	176.3	176.3	176.3	172.3	176.0	167.1	167.1
2′	33.9	33.9	33.9	33.9	42.8	41.1	127.5	127.5
3′	18.0	18.0	18.0	18.0	25.3	26.5	138.1	138.0
4′	17.9	17.9	17.9	17.9	21.4	15.7	19.3	19.5
5′					21.5	10.8	14.6	14.6
1′′	168.0	173.4	177.4	177.4	173.4	173.4	173.4	168.0
2′′	128.0	43.1	34.1	136.6	43.1	43.1	43.1	128.0
3′′	137.4	25.4	18.5	129.4	25.3	25.4	25.4	137.4
4′′	19.5	21.4	17.8	17.1	21.4	21.4	21.4	19.2
5′′	14.6	21.4			21.4	21.5	21.4	14.6

^a^ Measured at 150 MHz in methanol-*d*_4_; ^b^ Measured at 125MHz in methanol-*d*_4_.

**Table 3 molecules-27-05991-t003:** Cytotoxicity of compounds **1**–**8**.

Compound	IC_50_ (μM)		
	Hep G2	MCF-7	A549
**1**	>40	>40	>40
**2**	>40	>40	>40
**3**	>40	>40	>40
**4**	>40	37.32 ± 0.24	>50
**5**	>40	>40	>40
**6**	>40	>40	>40
**7**	>40	>40	>40
**8**	>40	>40	>40
*cis*-platin	16.20 ± 0.24	22.80 ± 0.83	27.07 ± 0.15

Values were mean ± SD, *cis*-platin, positive control.

## Data Availability

All data and figures in this study are openly available.

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
