# Peer review of "New 11-Methoxymethylgermacranolides from the Whole Plant of Carpesium divaricatum"

_molecules, 2022, doi:10.3390/molecules27185991_

Round 1

Reviewer 1 Report

Minor corrections needed. Please review the NMR proton integrations. PDF with corrections included.

Author Response

Minor corrections needed. Please review the NMR proton integrations. PDF with corrections included.

Response: Thanks a lot. We have checked the NMR proton integrations and have made corrections according to the suggestions marked the PDF.

The “o” of 4.53 (1H, o, H-8) means overlapped, and the annotation are made in the Table 1.

Meanwhile, the reference for cardivarolide I is reference 14 and the basic structure of cardivarolides are same as subtypes IV. We have added some description in the text as follows:  

Subtypes IV (the basic structure of cardivarolides) and V have similar skeletons except for the presence of a methoxymethyl group instead of the Δ11,13 exocyclic methylene group in the 5-membered ring [12–13] (Figure 1).

Reviewer 2 Report

The authors reported eight new 11-methoxymethylgermacranolides (18) isolated from the ethanol extract of the whole plant of Carpesium divaricatum. The structures of the new compounds were determined by detailed spectroscopic analysis, ECD comparison or calculation and X-ray crystallographic analysis. Compounds 1–8 represent a new subtype V of germacranolide. Compound 4 exhibited weak cytotoxicity against MCF-7 cells. Compound 8 decreased PGE2 productions in LPS-induced RAW 264.7 cells. The research is interesting and the manuscript was well written. I recommend it to be published in Molecules after minor revision.

1.      Line 51, change 3.35 (1H, s H-16) to 3.35 (3H, s H-16).

2.      Line 52 and line 55, check the signals of 5.44 (H-4), 4.52 (H-6), 6.12 (H-3”), which is a little different from those in table 1.

3.      All circular dichroism (CD) should be electronic circular dichroism (ECD).

Author Response

Reviewer #2: The authors reported eight new 11-methoxymethylgermacranolides (18) isolated from the ethanol extract of the whole plant of Carpesium divaricatum. The structures of the new compounds were determined by detailed spectroscopic analysis, ECD comparison or calculation and X-ray crystallographic analysis. Compounds 1–8 represent a new subtype V of germacranolide. Compound exhibited weak cytotoxicity against MCF-7 cells. Compound decreased PGE2 productions in LPS-induced RAW 264.7 cells. The research is interesting and the manuscript was well written. I recommend it to be published in Molecules after minor revision.

  1. Line 51, change 3.35 (1H, s H-16) to 3.35 (3H, s H-16).

Response: We are sorry for this mistake. We have changed 3.35 (1H, s H-16) to 3.35 (3H, s H-16).

  1. Line 52 and line 55, check the signals of 5.44 (H-4), 4.52 (H-6), 6.12 (H-3”), which is a little different from those in table 1.

Response: We are sorry for this mistake. We have checked the signals of you mentioned and have modified the errors.

  1. All circular dichroism (CD) should be electronic circular dichroism (ECD).

Response: Thanks for your suggestions. We have modified all “CD” to ECD in this paper and in the supporting information.

Reviewer 3 Report

This paper has reported the isolation and structural elucidation of eight new 11-methoxymethylgermacranolides isolated from the ethanol extract of the whole plant of Carpesium divaricatum. The absolute configuration of these isolates was determined by ECD calculations and X-ray crystallographic analysis. The anti-inflammatory activities of all isolated compounds were evaluated. It is informative to readers, and I highly recommend this manuscript to be published in this journal after a minor revision.

Comments:

1: Figure 1, 2, 3, and 4 should be rearranged to make it easier for readers

2: Please expand the abbreviations in the 3.1 section (HRESIMS, HPLC, TLC …)

3: The chemical shift values in the 1H and 13C spectrum of compounds 1, 2, 3, 5, 6, and 8 (supplementary material) is hard to read. Please improve the resolution of these figures.

4: Line 166: “when teated with compounds …” ?

Author Response

Reviewer #3: This paper has reported the isolation and structural elucidation of eight new 11-methoxymethylgermacranolides isolated from the ethanol extract of the whole plant of Carpesium divaricatum. The absolute configuration of these isolates was determined by ECD calculations and X-ray crystallographic analysis. The anti-inflammatory activities of all isolated compounds were evaluated. It is informative to readers, and I highly recommend this manuscript to be published in this journal after a minor revision.

Comments:

  1. Figure 1, 2, 3, and 4 should be rearranged to make it easier for readers

Response: Thanks for your suggestions. We have rearranged the position of figure 1, 2, 3, and 4.

  1. Please expand the abbreviations in the 3.1 section (HRESIMS, HPLC, TLC …)

Response: Thanks for your suggestions. We have expanded the abbreviations you mentioned in the 3.1 section.

  1. The chemical shift values in the 1H and 13C spectrum of compounds 1, 2, 3, 5, 6, and 8 (supplementary material) is hard to read. Please improve the resolution of these figures.

Response: We have provided the high resolution of all spectra of compounds 1, 2, 3, 5, 6, and 8.

  1. Line 166: “when teated with compounds …” ?

Response: We are sorry for this mistake. We have made a correction and modified “teated” to “treated”.
